# Prognostic value of high-sensitivity cardiac troponin I in heart failure patients with mid-range and reduced ejection fraction

Petr Lokaj[1,2], Jindrich Spinar[2,3], Lenka Spinarova[2,3], Filip Malek[4], Ondrej Ludka[1,2], Jan Krejci[2,3], Petr Ostadal[4], Dagmar Vondrakova[4], Karel Labr[2,3], Monika Spinarova[2,3], Monika Pavkova Goldbergova[5], Marie Miklikova[1,2], Katerina Helanova[1,2], Ilona Parenicova[6], Vladimir Jakubo[1,2], Klara Benesova[7], Roman Miklik[8], Jiri Jarkovsky[7], Tomas Ondrus[1,2☯ *], Jiri Parenica[1,2☯]

1 Department of Internal Medicine and Cardiology, University Hospital Brno, Brno, Czech Republic, 2 Faculty of Medicine, Masaryk University, Brno, Czech Republic, 3 First Department of Internal Medicine, Cardiology and Angiology, St Anne's University Hospital Brno, Brno, Czech Republic, 4 Department of Cardiology, Hospital Na Homolce, Prague, Czech Republic, 5 Department of Pathophysiology, Faculty of Medicine, Masaryk University, Brno, Czech Republic, 6 Center of Cardiovascular Surgery and Transplantations, Brno, Czech Republic, 7 Institute of Biostatistics and Analyses, Faculty of Medicine, Masaryk University, Brno, Czech Republic, 8 Department of Cardiology, University Hospital Plzen, Plzen, Czech Republic

☯ These authors contributed equally to this work.
* ondrus.tomas@fnbrno.cz

**Data Availability Statement:** All relevant data are within the mansucript and its Supporting information files.

## Abstract

### Background

The identification of high-risk heart failure (HF) patients makes it possible to intensify their treatment. Our aim was to determine the prognostic value of a newly developed, high-sensitivity troponin I assay (Atellica®, Siemens Healthcare Diagnostics) for patients with HF with reduced ejection fraction (HFrEF; LVEF < 40%) and HF with mid-range EF (HFmrEF) (LVEF 40%–49%).

### Methods and results

A total of 520 patients with HFrEF and HFmrEF were enrolled in this study. Two-year all-cause mortality, heart transplantation, and/or left ventricular assist device implantation were defined as the primary endpoints (EP). A logistic regression analysis was used for the identification of predictors and development of multivariable models. The EP occurred in 14% of the patients, and these patients had higher NT-proBNP (1,950 vs. 518 ng/l; p < 0.001) and hs-cTnI (34 vs. 17 ng/l, p < 0.001) levels. C-statistics demonstrated that the optimal cut-off value for the hs-cTnI level was 17 ng/l (AUC 0.658, p < 0.001). Described by the AUC, the discriminatory power of the multivariable model (NYHA > II, NT-proBNP, hs-cTnI and urea) was 0.823 (p < 0.001). Including heart failure hospitalization as the component of the combined secondary endpoint leads to a diminished predictive power of increased hs-cTnI.

**Funding:** This work was supported by the Ministry of Health of the Czech Republic as part of the project Conceptual Development of Research Organisation (University Hospital Brno, project 65269705) and a project of the Czech Health Research Council of the Ministry of Health of the Czech Republic (NV18-09-00146). The funders had no role in study design, data collection and analysis, decision to publish, or preparation of the manuscript.

**Competing interests:** The authors have declared that no competing interests exist.

## Conclusion

hs-cTnI levels $\geq$ 17 ng/l represent an independent increased risk of an adverse prognosis for patients with HFrEF and HFmrEF. Determining a patient's hs-cTnI level adds prognostic value to NT-proBNP and clinical parameters.

## Introduction

The prognosis of patients with chronic heart failure is rather poor; the 3-year all-cause mortality is approximately 35% [1]. Various prognostic scoring systems can identify the highest-risk patients and warn their physicians that, in accordance with clinical practice guidelines, further diagnostic tests need to be done, pharmacotherapies should be modified, non-pharmacological treatments should be considered, or, as a last resort, left ventricular assist device (LVAD) implantation or heart transplantation (HTX) should be considered [2]. There are a limited number of models for stable patients with chronic heart failure that evaluate cohorts of patients treated according to the current guidelines and, apart from clinical and classic laboratory parameters, also use natriuretic peptides and other novel biomarkers. In addition, most models are based on either overall mortality or cardiovascular mortality as the monitored endpoint, thus omitting a particularly important component of terminal heart failure treatment, i.e. LVAD implantation or heart transplantation, both of which can alter the natural course of the disease and improve the patient's prognosis. Importantly, the discriminatory power of the currently used models tends to be lower when monitoring hospitalisations for heart failure [3]. For these reasons, we used the overall mortality, LVAD implantation, and/or HTX as the primary combined endpoints in our study. The secondary combined endpoints were defined as combination of primary endpoints and acute heart failure hospitalization.

Establishing the natriuretic peptide levels is integral to providing care for heart failure patients, as these peptides are the most important prognostic markers [4, 5]. High-sensitivity cardiac troponin is another cardiac biomarker that is readily available online and in standard clinical laboratories. A recently published meta-analysis showed that hs-cTnT is an independent predictor of heart failure patients' prognoses [6]; however, there is only a limited amount of information on the prognostic significance of hs-cTnI.

The aim of our analysis was to describe the prognostic significance of a novel high-sensitivity cardiac troponin I assay (Atellica® IM high-sensitivity troponin I, Siemens Healthineers) in a cohort of patients with stable systolic chronic heart failure (reduced or mid-range ejection fraction) and evaluate its contribution to NT-proBNP levels and clinical parameters.

## Methods

This study protocol complies with the regulations set forth in the Declaration of Helsinki and was approved by the Ethics Committee of the University Hospital Brno (Brno, Czech Republic). Written informed consent was obtained from each of the patients before they began their participation in the study. The study protocol was described in our previous study [4]; in short, a total of 1,088 patients were prospectively recruited from November 2014 to November 2015, and 520 of them were evaluated in the current sub-study. The primary endpoint, i.e., the two-year prognosis for all-cause mortality, heart transplantation, and/or left ventricular assist device (LVAD) implantation, was evaluated up to November 2017. The secondary combined endpoints were defined as combination of the two-year all-cause mortality, heart

transplantation, LVAD implantation and/or acute heart failure hospitalization. The patients were followed up prospectively at outpatient departments, and the mortality rates were verified using the centralised database of the Czech Republic Ministry of Health. The monitored data from the patients were gathered at the end of the two-year follow-up period.

## Study population

Patients with a stable form of either heart failure with reduced ejection fraction (HFrEF) (LVEF < 40%) or heart failure with mid-range ejection fraction (HFmrEF) (LVEF 40%–49%) were eligible for inclusion in this study. The cohort included patients who were followed up and treated in outpatient clinics of cardiology departments (where authors of this study work) for stable chronic heart failure. All three cardiology departments provide specialised care for heart failure patients. In their medical history, all patients included in the study had an attack of heart failure with elevated natriuretic peptides and a reaction to heart failure treatment. Echocardiography was performed for all patients by experienced physicians working in echocardiography laboratories of cardiology departments. Other structural and/or functional abnormalities related to the patients' heart failure were found, including LVEF values ranging between 40% and 49%: left ventricular hypertrophy (an interventricular septum $\geq$ 11 mm or a left ventricular mass index $\geq$ 115 g/m$^2$ for the men and $\geq$ 95 g/m$^2$ for the women), left atrial enlargement (a left atrial volume index > 34 ml/m$^2$) and/or diastolic dysfunction (E/e' $\geq$ 13 and mean e' < 9 cm/s). A current NT-proBNP level < 125 pg/mL was not among the exclusion criteria because all the patients had increased NT-proBNP levels in their medical history. The exclusion criteria were the following: not signing the informed consent, signs and symptoms of acute decompensation from heart failure, and conditions other than heart failure that would likely hinder the patients' mid-term prognosis (e.g., advanced cancer, severe dementia, etc.). The final decision about the diagnosis of chronic heart failure and the enrolment of a specific patient in the study was done by a cardiologist experienced in care for heart failure patients.

## Laboratory methods

The patient plasma NT-ProBNP levels were analysed using the Cobas E411 NT-proBNP electrochemiluminescence immunoassay Kit (Elecsys proBNP II, Roche Diagnostics, Indianapolis, IN, USA). The limit of blank (LoB) was 3 pg/mL, the limit of detection (LoD) was 5 pg/mL, the measuring range was 5–35,000 pg/mL, the functional sensitivity (the lowest analyte concentration that can be reproducibly measured with an intermediate precision CV of 20%) was 50 pg/mL, and the cut-off value was 125 pg/mL.

The plasma hs-cTnI levels were analysed using the Atellica® IM High sensitivity Troponin I assay (Atellica® IM TnIH, Siemens Healthcare Diagnostics Inc., Tarrytown, NY, USA), run on an Atellica® IM Analyser. The LoB of the Atellica® IM TnIH assay was 0.50 ng/L. The observed LoD ranged from 1.13 to 1.53 ng/L across three reagent lots and two matrices (serum and lithium heparin plasma). The limit of quantitation (LoQ) of the Atellica® IM TnIH assay was 2.50 ng/l. The Atellica® IM TnIH assay provides results from 2.50 to 25,000.00 ng/L. The lower end of the measuring interval is defined by the LoQ. The Atellica® IM TnIH assay is a 3-site sandwich immunoassay using direct chemiluminometric technology. The solid phase reagent is based on magnetic latex particles conjugated with streptavidin with two bound biotinylated capture monoclonal antibodies, each recognising a unique cTnI epitope. The 99th percentile upper reference limit for healthy individuals was 34 ng/L for women and 53 ng/l for men. The total imprecision (CV) at the 99th percentile value of 45.20 pg/mL (ng/L) was below 10%.

## Statistical methods

Standard descriptive statistics were applied in the analysis: the continuous variables were described as the mean ± SD and the median (5th percentile; 95th percentile), whereas the categorical variables were characterised by absolute and relative frequencies. The statistical significance of the differences among the groups of patients was analysed using the Mann-Whitney U test for continuous variables and the Fisher's exact test for categorical variables. The contribution of the hs-cTnI biomarker to the NT-proBNP levels and clinical model was evaluated according to previously published recommendations [7, 8]. A logistic regression was adopted for the identification of predictors and development of multivariable models for scoring systems and biomarkers. The models were evaluated using a flexible calibration curve [9], C statistics, and a reclassification analysis of the model results. The analysis was completed using SPSS 24.0.0.1 (IBM Corporation, 2016) and R 3.5.1, with the PredictABEL and rms package.

## Results

A total of 520 patients were evaluated. During the two-year follow-up, the primary endpoint occurred in 73 patients (14.0%). Of these patients, 55 died (without a previous LVAD implantation/HTX), 3 underwent LVAD implantations, 3 underwent LVAD implantations followed by HTX, 2 underwent LVAD implantation and later died, and 10 underwent HTX only. Acute heart failure hospitalization occurred in 74 patients (14.2%) of whom 30 patients (5.7%) experienced a further study endpoint and 44 (8.5%) heart failure hospitalization only. The secondary combined endpoint occurred in 117 patients (S1 Table). In general, the primary endpoint occurred more frequently in the patients who had a lower diastolic blood pressure (76 ± 9 vs. 81 ± 11 mmHg; p = 0.025), lower LVEF (27 ± 9 vs. 32 ± 9%; p < 0.001), and severe dyspnoea, i.e., those classified as NYHA III–IV (37.0% vs. 15.4%; p = 0.001). These patients also had higher hs-cTnI (34 vs. 17 ng/L; p < 0.001), NT-proBNP (1,950 vs. 518 ng/L; p < 0.001), and urea (8 vs. 6 mmol/L; p < 0.001) levels, and lower haemoglobin (135 vs. 145 g/L; p = 0.002) levels. The primary endpoint also occurred more frequently in the patients with a higher dose of furosemide ($\geq$ 40 mg/day), but their heart failure treatment was otherwise comparable to the treatment given the patients without the primary endpoint (Table 1). A comparison of the characteristics of patients with/without the secondary endpoint is provided in S2 Table. Both groups differed in similar parameters as in the primary endpoint analysis (only statistically higher systolic BP was found in patients with the secondary endpoint, diastolic BP was comparable between groups). The patients with the secondary endpoint were also found to have significantly higher hs-cTnI levels (31 vs 16 ng/L; p < 0.001).

### Relationship of hs-cTnI to clinical and laboratory parameters

The hs-cTnI levels differed significantly among the groups of patients categorized by age, LVEF, and renal function (expressed by the eGFR). In contrast, there were no significant differences found among the groups classified by sex, BMI, the ischaemic aetiology of the patients' heart failure, hypertension, atrial fibrillation, diabetes mellitus, chronic obstructive pulmonary disease, or lower extremity peripheral artery disease (S3 Table). According to the Spearman's rank correlation coefficient, the hs-cTnI levels were most strongly linked to the NT-proBNP levels (r = 0.439; p < 0.001) and renal function (the correlation coefficients for creatinine, urea, and eGFR were r = 0.245, 0.241, and -0.240, respectively, with p < 0.001 for all three parameters). The link was somewhat weaker when LVEF (r = -0.208), heart rate (r = 0.198), and age (r = 0.186) were considered (p < 0.001 for all three parameters) (S4 Table).

**Table 1. Basic characteristics of the patients by occurance of the primary endpoint.**

| Parameter | Total (N = 520) | Without endpoint (n = 447) | With endpoint (n = 73) | P-value |
|---|---|---|---|---|
| **Basic characteristics** | | | | |
| Sex–male | 419 (80.6%) | 360 (80.5%) | 59 (80.8%) | NS |
| Age | 65 ± 12 | 65 ± 12 | 67 ± 13 | NS |
| BMI | 29 ± 5 | 29 ± 5 | 29 ± 5 | NS |
| SBP [mmHg] | 127 ± 15 | 128 ± 15 | 122 ± 15 | NS |
| DBP [mmHg] | 80 ± 10 | 81 ± 11 | 76 ± 9 | **0.025** |
| Heart rate [min⁻¹] | 73 ± 13 | 72 ± 13 | 76 ± 12 | NS |
| LVEF [%] | 32 ± 9 | 32 ± 9 | 27 ± 9 | **< 0.001** |
| Ischaemic aetiology of HF | 283 (54.4%) | 243 (54.4%) | 40 (54.8%) | NS |
| Hypertension | 344 (66.2%) | 290 (64.9%) | 54 (74.0%) | NS |
| Atrial fibrillation | 173 (33.3%) | 148 (33.1%) | 25 (34.2%) | NS |
| Diabetes mellitus | 205 (39.4%) | 167 (37.4%) | 38 (52.1%) | NS |
| COPD | 80 (15.4%) | 62 (13.9%) | 18 (24.7%) | NS |
| Lower extremity peripheral artery disease | 49 (9.4%) | 38 (8.5%) | 11 (15.1%) | NS |
| Smoking | | | | NS |
| Non-smoker | 298 (57.3%) | 257 (57.5%) | 41 (56.2%) | |
| Smoker | 55 (10.6%) | 49 (11.0%) | 6 (8.2%) | |
| Ex-smoker | 167 (32.1%) | 141 (31.5%) | 26 (35.6%) | |
| NYHA classification | | | | **0.001** |
| 1 | 75 (14.4%) | 71 (15.9%) | 4 (5.5%) | |
| 2 | 349 (67.1%) | 307 (68.7%) | 42 (57.5%) | |
| 3–4 | 96 (18.5%) | 69 (15.4%) | 27 (37.0%) | |
| **Laboratory results** | | | | |
| hs-cTnI [ng/l] | 19 (4; 339) | 17 (4; 280) | 34 (7; 406) | **< 0.001** |
| NT-proBNP [ng/l] | 690 (44; 6,038) | 518 (39; 4,514) | 1,950 (335; 16,768) | **< 0.001** |
| Haemoglobin [g/l] | 144 (114; 167) | 145 (115; 169) | 135 (105; 160) | **0.002** |
| Natrium [mmol/l] | 141 (135; 146) | 141 (135; 146) | 140 (133; 145) | NS |
| Urea [mmol/l] | 6 (4; 15) | 6 (4; 13) | 8 (4; 22) | **< 0.001** |
| Uric acid [µmol/l] | 397 (234; 592) | 391 (236; 583) | 436 (221; 626) | NS |
| Creatinine [µmol/l] | 95 (66; 176) | 94 (67; 169) | 103 (62; 261) | NS |
| eGFR [ml/min/1.73 m²] | 70 (29; 103) | 70 (32; 103) | 62 (18; 100) | NS |
| **Medication** | | | | |
| ACEI/ARB | 464 (89.2%) | 403 (90.2%) | 61 (83.6%) | NS |
| Beta-blockers | 485 (93.3%) | 419 (93.7%) | 66 (90.4%) | NS |
| Furosemide ≥ 40 mg/day | 296 (56.9%) | 239 (53.5%) | 57 (78.1%) | **0.002** |
| Spironolactone/eplerenone | 334 (64.2%) | 282 (63.1%) | 52 (71.2%) | NS |

The categorical variables are characterised by absolute and relative frequencies. The continuous basic characteristics are described as the mean ± SD, and laboratory results are described as the median (5th–95th percentile). BMI, body mass index; COPD, chronic obstructive pulmonary disease; DBP, diastolic blood pressure; HF, heart failure; LVEF, left ventricular ejection fraction; SBP, systolic blood pressure; eGFR, estimated glomerular filtration rate (using the CKD-EPI equation).
The p-value of the Fisher's exact test for categorical variables and the p-value of the Mann-Whitney U test are shown with the Bonferroni correction applied.

## The prognostic value of hs-cTnI

Based on the ROC analysis (Table 2), we determined that 17 ng/L is the optimum cut-off value for the hs-cTnI level for predicting the monitored primary endpoint. The area under the curve (AUC) in this model was 0.658 ($p < 0.001$) (S1A Fig). The ROC analysis of hs-cTnI as a continuous parameter did not demonstrate any statistical significance. Fig 1 shows the distribution

**Table 2. NT-proBNP [ng/l] and hs-cTnI [ng/l] as predictors of the primary endpoint in the logistic regression models (i.e., the two-year prognosis in terms of all-cause mortality, heart transplantation and left ventricular assist device [LVAD] implantation).**

| Predictor | | OR (95% CI) | P | AUC (95% CI) |
|---|---|---|---|---|
| **Univariable models:** | | | | |
| NT-proBNP–categorical | 1-category increase* | 1.92 (1.56; 2.36) | < **0.001** | 0.749 (0.695; 0.803) |
| hs-cTnI | 100-unit increase | 1.04 (0.98; 1.12) | 0.214 | 0.663 (0.601; 0.725) |
| hs-cTnI–binary | ≥ 17 (ref. < 17) | 4.35 (2.36; 8.01) | < **0.001** | 0.658 (0.596; 0.721) |
| **Multivariable model (combination of NT-proBNP and hs-cTnI):** | | | | |
| NT-proBNP–categorical | 1-category increase* | 1.76 (1.42; 2.18) | < **0.001** | 0.767 (0.715; 0.819) |
| hs-cTnI [ng/l]–binary | ≥ 17 (ref. < 17) | 2.30 (1.20; 4.41) | **0.012** | |

*Categories of NT-proBNP: < 100 / 100–249 / 250–499 / 500–999 / 1,000–1,999 / ≥ 2,000.

of the hs-cTnI levels and the primary endpoint occurrence in the individual categories, according to the increasing levels of hs-cTnI. Patients with hs-cTnI levels < 17 ng/L are at a low risk of endpoint occurrence, whereas those with hs-cTnI levels ≥ 17 ng/L are at a high risk. It should be noted, however, that a further increase in troponin levels does not translate into a significant increase in the risk of primary endpoint occurrence. Patients with hs-cTnI levels < 17 ng/L accounted for 46.4% of all the patients. The monitored primary endpoint did not occur in any of the patients with hs-cTnI levels lower than the LoB (< 3 ng/L), but these patients only accounted for 2.9% of the total patients.

Similarly, based on the ROC analysis, we determined that 17 ng/L is also the optimum cut-off value for the hs-cTnI level for predicting the secondary endpoint. The area under the curve (AUC) in this model was 0.647 (p < 0.001) (S1B Fig).

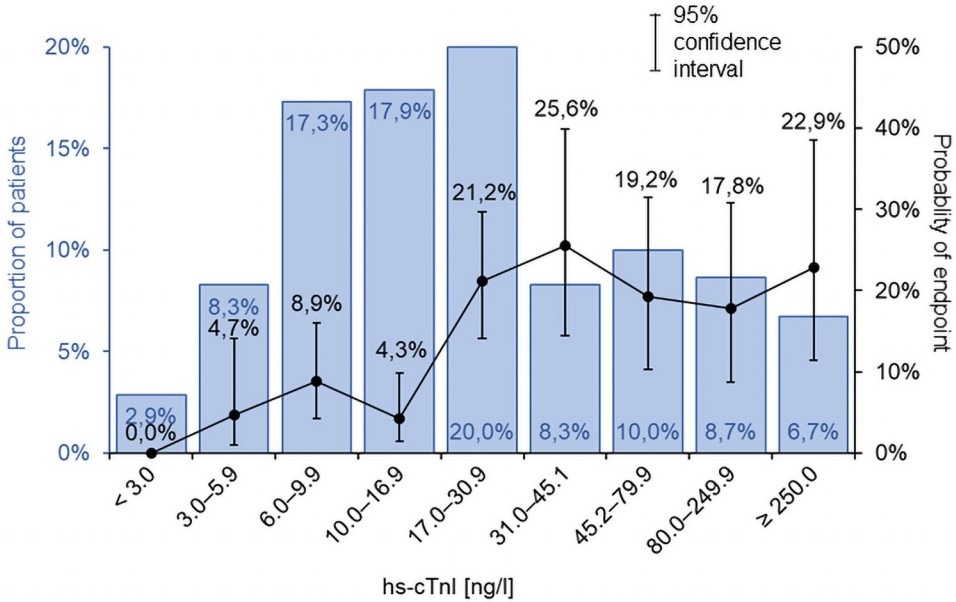

**Fig 1. hs-cTnI distribution in patients with chronic heart failure and the associated 2-year occurrence of the combined endpoints (death, HTX or LVAD).**

**Table 3. Patient characteristics as predictors of the primary endpoint in the univariable logistic regression models (i.e., the two-year prognosis in terms of all-cause mortality, heart transplantation and left ventricular assist device [LVAD] implantation).**

| Predictor | | OR (95% CI) | P |
|---|---|---|---|
| Sex | Men (ref. women) | 1.02 (0.54; 1.91) | 0.954 |
| Age | ≥ 65 (ref. < 65) | 2.01 (1.19; 3.41) | **0.009** |
| BMI | ≥ 30 (ref. < 30) | 1.15 (0.69; 1.91) | 0.585 |
| SBP [mmHg] | ≥ 110 (ref. < 110) | 0.40 (0.20; 0.81) | **0.010** |
| DBP [mmHg] | ≥ 80 (ref. < 80) | 0.54 (0.33; 0.89) | **0.015** |
| Heart rate [min$^{-1}$] | ≥ 75 (ref. < 75) | 1.95 (1.18; 3.23) | **0.009** |
| LVEF [%] | ≥ 35 (ref. < 35) | 0.36 (0.21; 0.65) | **< 0.001** |
| Ischaemic aetiology of HF | Yes (ref. no) | 1.02 (0.62; 1.67) | 0.945 |
| Hypertension | Yes (ref. no) | 1.54 (0.88; 2.69) | 0.130 |
| Atrial fibrillation | Yes (ref. no) | 1.05 (0.62; 1.77) | 0.848 |
| Diabetes mellitus | Yes (ref. no) | 1.82 (1.11; 2.99) | **0.018** |
| COPD | Yes (ref. no) | 2.03 (1.12; 3.69) | **0.020** |
| Lower extremity peripheral artery disease | Yes (ref. no) | 1.91 (0.93; 3.93) | 0.079 |
| NYHA classification | > 2 (ref. ≤ 2) | 3.78 (2.27; 6.29) | **< 0.001** |
| Anaemia | Yes (ref. no) | 2.36 (1.35; 4.10) | **0.002** |
| Natrium [mmol/l] | ≥ 135 (ref. < 135) | 0.35 (0.14; 0.88) | **0.026** |
| Urea [mmol/l] | ≥ 6 (ref. < 6) | 4.19 (2.33; 7.52) | **< 0.001** |
| Uric acid [μmol/l] | ≥ 500 (ref. < 500) | 2.82 (1.61; 4.96) | **< 0.001** |
| Creatinine [μmol/l] | ≥ 100 (ref. < 100) | 2.09 (1.27; 3.44) | **0.004** |
| eGFR [ml/min/1.73 m$^2$] | ≥ 60 (ref. < 60) | 0.55 (0.33; 0.91) | **0.020** |
| ACEI/ARB | Yes (ref. no) | 0.56 (0.28; 1.11) | 0.096 |
| Beta-blockers | Yes (ref. no) | 0.63 (0.26; 1.50) | 0.297 |
| Furosemide ≥ 40 mg/day | ≥ 40 mg/day (ref. < 40) | 3.10 (1.73; 5.56) | **< 0.001** |
| Spironolactone/eplerenone | Yes (ref. no) | 1.45 (0.84; 2.49) | 0.180 |

BMI, body mass index; COPD, chronic obstructive pulmonary disease; DBP, diastolic blood pressure; HF, heart failure; LVEF, left ventricular ejection fraction; SBP, systolic blood pressure; eGFR, estimated glomerular filtration rate (using the CKD-EPI equation).

## Multivariable prognostic models for primary endpoint

The AUC value for the prediction of the primary endpoint based on the ROC analysis of the NT-proBNP levels was 0.749 (p < 0.001) for the categorical model. The predictive value of the NT-proBNP plus hs-cTnI model was 0.767 (p < 0.001), according to the AUC (Table 2).

Based on previously published prognostic models, we developed a multivariable model by selecting comorbidities and clinical and laboratory parameters that have been shown to influence the prognosis of chronic heart failure patients [3, 4, 10]. Using a univariable logistic regression, we identified parameters that were significantly related to the monitored primary endpoint (Table 3). These parameters, together with the NT-proBNP and hs-cTnI levels, were used as inputs for the multivariable logistic regression. The significance of the individual variables selected for the multivariable model is shown in S1 Fig. The final multivariable model, which included NT-proBNP levels, NYHA (> 2), urea, and hs-cTnI levels (≥ 17 ng/l), was created using a backward stepwise algorithm (Table 4). The excellent discriminatory power of the model to distinguish between patients with good and poor prognoses, assessed by the ROC analysis and expressed by the AUC, was 0.823 (p < 0.001) (S5 Table). Adding other parameters

**Table 4. The multivariable logistic regression model using a backward stepwise algorithm for the selection of independent predictors of the primary endpoint (i.e., the two-year prognosis in terms of all-cause mortality, heart transplantation and left ventricular assist device [LVAD] implantation).**

| Predictor | | OR (95% CI) | P |
|---|---|---|---|
| NYHA | > 2 (ref. ≤ 2) | 3.24 (1.84; 5.68) | **< 0.001** |
| hs-cTnI [ng/l] | ≥ 17 (ref. < 17) | 2.13 (1.08; 4.18) | **0.028** |
| NT-proBNP [ng/l] | 1-category increase* | 1.60 (1.29; 2.00) | **< 0.001** |
| Urea [mmol/l] | 1-category increase** | 2.18 (1.47; 3.24) | **< 0.001** |

*Categories of NT-proBNP: < 100 / 100–249 / 250–499 / 500–999 / 1,000–1,999 / ≥ 2,000.

**Categories of urea: < 6 / 6–9.9 / ≥ 10.

to this model did not improve its discriminatory power. After performing the 10-fold cross-validation to validate the results, the average AUC of 0.804 (in the range 0.689–0.892) was obtained. The calibration of the model was assessed by a flexible calibration curve, which confirmed that the predictive model is well calibrated (S3 Fig).

Our previously published multivariable model for predicting the monitored two-year endpoint (overall mortality, HTX, or LVAD implantation) is based solely on clinical parameters and NT-proBNP levels [4]. This model includes the following independent parameters: older age, advanced heart failure (NYHA III+IV), anaemia, hyponatraemia, hyperuricaemia, and a higher dose of furosemide (> 40 mg daily). According to the ROC statistics and expressed by the AUC, the predictive power of this model, applied to the population of 520 patients included in this analysis, was 0.777 (0.722; 0.832) ($p < 0.001$).

A reclassification analysis confirmed the improvement of the predictive power of the model that combined the NT-proBNP and hs-cTnI levels, urea, and NYHA, as opposed to the model based solely on the NT-proBNP level and clinical parameters: the category-free net reclassification improvement was 0.473 (0.224; 0.723) ($p < 0.001$) and the integrated discrimination improvement was 0.054 (0.014; 0.094) ($p = 0.008$).

The final model was visualised using a nomogram (Fig 2), which makes it possible to establish the risk for each patient individually, based on their NT-proBNP and hs-cTnI levels, urea, and NYHA.

## Multivariable prognostic model for secondary endpoint

To create a multivariable model for secondary endpoint prediction, we proceeded similarly as in the primary endpoint analysis. Using a univariable logistic regression, we identified parameters that were significantly related to the occurrence of the secondary endpoint (age, systolic blood pressure, diastolic blood pressure, left ventricular ejection fraction, hypertension, diabetes mellitus, chronic obstructive pulmonary disease, NYHA classification, anaemia, natrium, urea, uric acid, estimated glomerular filtration rate, use of ACE inhibitors/angiotensin II receptor blockers, beta-blockers and higher dose of furosemide; the data are not presented). These parameters, together with NT-proBNP and hs-cTnI levels, were used as inputs for the multivariable logistic regression. The final multivariable model was created using a backward stepwise algorithm that included NYHA (> 2), NT-proBNP and urea levels. The level of hs-cTnI was not found to be an independent predictor of the secondary endpoint (S6 Table). The discriminatory power of the model to distinguish between patients with good and poor prognosis, assessed by the ROC analysis and expressed by the AUC, was 0.802 ($p < 0.001$).

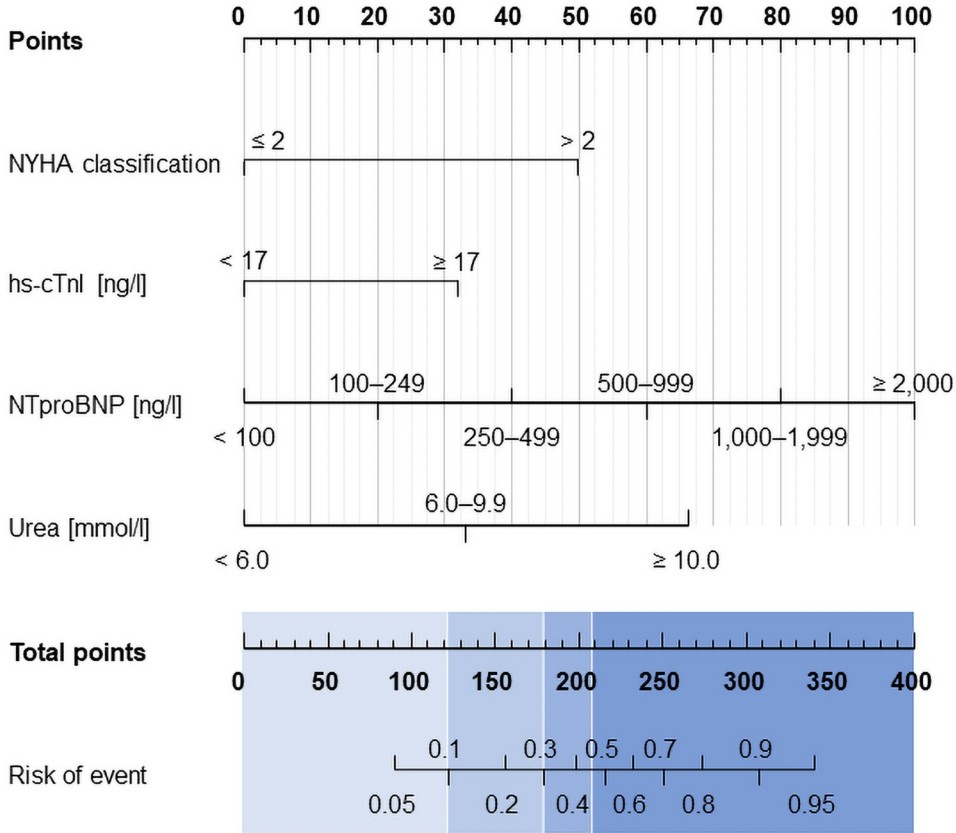

**Fig 2. The nomogram of the proposed risk score.**

## Comparison of HFrEF and HFmrEF patient subpopulations

In our opinion, it is interesting to highlight some differences between the HFrEF and HFmrEF patient subpopulations. Men outnumber women in both groups; however, the proportion of women is higher in the HFmrEF group than in the HFrEF group (30% and 16.2%, respectively). Furthermore, HFmrEF patients are older, have a higher systolic blood pressure, and their heart failure is of ischaemic aetiology much more frequently. Overall, patients in the HFmrEF group are less frequently classified as NYHA III-IV, have lower median levels of hs-cTnI, NT-proBNP and uric acid, and are less frequently treated with furosemide $\geq$ 40 mg/day and with mineralocorticoid receptor antagonists (S7 Table). According to Kaplan-Meier curves, the two-year survival without the monitored endpoint is 90.7 (85.1–96.3) in the HRmrEF group, and 82.7 (78.7–86.7) in the HRrEF group. According to the ROC analysis of our multivariable model, the area under the curve (AUC) is 0.860 (N = 120) for HRmrEF and 0.813 (N = 400) for HRrEF. The cut-off values for hs-TnI for HRrEF and HRmrEF patient subpopulations were determined to be 17 ng/l (N = 400) and ideally 14 ng/l (N = 120), respectively; but even 17 ng/l is a very good cut-off value for the HRmrEF group. In view of the fact that the HRmrEF group involves a markedly lower number of patients, we consider a single cut-off value to be adequate, namely 17 ng/l.

## Discussion

This is the first study to evaluate the prognostic significance of cardiac troponin I (Atellica$^{\circledR}$ IM High sensitivity Troponin I) in patients with chronic heart failure. This paper presents

three important results. First, the hs-cTnI cut-off value for predicting a poorer prognosis of HFrEF/HFmrEF patients, as determined by Atellica®, is 17 ng/L, and a further increase in the troponin level is not linked to a significant risk increase. Second, the hs-cTnI levels in chronic heart failure patients are linked to their age, LVEF, and renal function, and they correlate with the NT-proBNP levels. Third, although the predictive value of the hs-cTnI level by itself, as expressed by the C-statistics, is relatively low, adding the hs-cTnI level to the NT-proBNP level and clinical parameters significantly improves the identification of patients that are at a higher risk of a combined endpoint (death, heart transplantation or LVAD implantation). From both analytical and clinical points of view, the value of hs-cTnI under 17 ng/l also helps to identify low-risk patients. Nevertheless, the value of hs-cTnI does not increase the power of the model for secondary endpoint prediction that includes decompensated heart failure hospitalizations (on top of primary endpoint components).

## Clinical benefit of the model and its interpretation

Our model makes it possible to identify the highest-risk patients in the population of HFrEF/HFmrEF patients. This subgroup of patients should be followed up in specialised centres for chronic heart failure patients and, in accordance with current guidelines, re-examinations, and possibly intensification of pharmacological treatments (increasing the dose of diuretics/ACE inhibitors/sartans/beta-blockers; replacing ACE inhibitors/sartans with sacubitril/valsartan; adding spironolactone/eplerenone, ivabradine, SGLT2 inhibitors or digoxin) and/or non-pharmacological treatments (revascularisation, ICD/CRT implantation, valvular heart disease surgery), might be considered. As a last resort, LVAD implantation or putting the patient on a waiting list for heart transplantation may also need to be considered [2].

From a practical standpoint, it is reasonable to find the boundary beyond which patients are at risk. During the two-year follow-up, the primary endpoint occurred in 14% of the patients in our cohort. In the group of patients with advanced heart failure, for whom the benefit of LVAD implantation was not yet definite (INTERMACS profile 6–7), the two-year occurrence of death/orthotopic heart transplantation/LVAD implantation was 42%. In the group of patients with an INTERMACS profile of 4–5, for whom the benefit of LVAD implantation (as compared with the optimum pharmacotherapy) had been demonstrated, the two-year occurrence of the combined endpoint was 53% [11]. In contrast, the two-year overall mortality rate of patients with advanced heart failure who had undergone the implantation of a centrifugal-flow pump (HeartMate 3) was 11.9%, and the survival at two years, free of disabling stroke or reoperation to replace or remove a malfunctioning device, was 79.5% [12].

With the use of the suggested predictive model, patients can be divided into four groups: low-risk patients (up to 10%, with a score of up to 120 points), low-to-moderate-risk patients (11%–30%, with a score between 121 and 180 points), moderate-to-high-risk (30%–45%, with a score between 181 and 208 points), and high-risk patients (45%, with a score of 209 or more points). Patients with extremely low hs-cTnI levels ($< 3$ ng/L) have a very good prognosis.

Intensification of pharmacological/non-pharmacological treatments might be considered for the group of low-to-moderate-risk patients, whereas moderate-to-high-risk and high-risk patients should be followed up in centres specialised in heart failure.

## Comparison with other models

The main advantages of our model are as follows: (1) the use of a combined endpoint, where LVAD implantation and heart transplantation are monitored together with mortality and (2) the use of routinely available biomarkers. We have demonstrated the benefit of adding the hs-cTnI level to the NT-proBNP level and clinical parameters. The two-year mortality prediction

was published in the Seattle Heart Failure Model, with an AUC of 0.792 [13]. The NT-proBNP level was also used in a recently published model for a two-year prediction of a primary composite endpoint (cardiovascular death or hospitalisation for heart failure) (AUC 0.71), cardiovascular death (AUC 0.71), and all-cause death (0.70) [5]. The use of biomarkers, particularly natriuretic peptides, should be a standard procedure to establish the degree of risk in heart failure patients. The NT-proBNP level was the most significant prognostic factor, not only in our study, but also in an extensive cohort of patients in the PARADIGM-HF study [5]. Our results are in accordance with a recently published, and rather extensive, meta-analysis that evaluated the prognostic value of hs-cTnT for individual patients with chronic heart failure. An hs-cTnT level $\geq$ 18 ng/L, added to clinical parameters and the NT-proBNP level, was an independent predictor for a more than two-year prognosis of monitored endpoints (all-cause mortality, cardiovascular mortality, and cardiovascular hospitalisation) [6].

Our study showed that including an acute heart failure hospitalization as the component of the combined endpoint leads to a diminished predictive power of increased hs-cTnI value. Similar findings were demonstrated by previous analyses of risk models in patients with heart failure published by Rahimi et al. The discriminatory ability of the models for prediction of death appeared to be higher than that for prediction of death or acute heart failure hospitalization or prediction of hospitalizations alone [3].

## Limitations

Our study has several limitations. First, we only described a population of patients with mid-range and reduced ejection fractions; any patient with a preserved ejection fraction was excluded from the study. Due to the limited number of patients, we did not attempt to establish whether men and women might have different cut-off values for predicting a poor prognosis. Our model was not validated in an external cohort of patients. Additionally, our results respond to hs-cTnI measurements in chronic heart failure patients in a stable phase in outpatient clinics. And finally, population sampling might be influenced by a selection bias. This group of heart failure patients has been followed up in university centres; therefore, the patient cohort very probably was not a representative sample of heart failure patients with EF < 50% from across the Czech Republic. A study from the Netherlands, based on an extensive registry of heart failure patients, which was intended to described the real-world population with heart failure [14], revealed that about 77% of all patients had EF < 50%. Their mean age was 72 years (as compared to 65 years in our cohort) and men accounted for 64% only (as compared to 80,6% in our cohort).

## Conclusion

Hs-cTnI levels $\geq$ 17 ng/l represent the cut-off for predicting a poorer prognosis for HFrEF/HFmrEF patients. We have developed a simple model for the early identification of HFrEF/HFmrEF patients whose prognoses might be improved if an intensification of treatment is considered (based on the indication criteria). This treatment intensification could include putting the patients on a waiting list for heart transplantation or LVAD implantation.

## Supporting information

**S1 Fig. A: ROC curve for prediction of combined primary endpoint (death, HTX or LVAD) using hsTnI; B: ROC curve for prediction of combined secondary endpoint (death, HTX, LVAD and/or acute heart failure hospitalization) using hsTnI.**
(TIF)

**S2 Fig. Significance of the variables as measured using a partial Wald $\chi^2$ test minus the predictor degrees of freedom in the full model and the final model selected by a backward stepwise algorithm.**
(TIF)

**S3 Fig. Calibration curve of the multivariable logistic regression model.**
(TIF)

**S1 Table. Summary of event occurrence in the first two years of follow-up.**
(DOCX)

**S2 Table. Basic characteristics of the patients according to occurrence of secondary endpoint (i.e., the two-year prognosis in terms of all-cause mortality, heart transplantation, left ventricular assist device [LVAD] implantation, hospitalization for HF).**
(DOCX)

**S3 Table. Comparison of hs-cTnI between patient subgroups.**
(DOCX)

**S4 Table. Correlation between hs-cTnI and other parameters.**
(DOCX)

**S5 Table. Accuracy of the multivariable prediction model.**
(DOCX)

**S6 Table. The multivariable logistic regression model using a backward stepwise algorithm for the selection of independent predictors of the primary endpoint (i.e., the two-year prognosis in terms of all-cause mortality, heart transplantation, left ventricular assist device [LVAD] implantation, hospitalization for HF).**
(DOCX)

**S7 Table. Basic characteristics of patients with HFrEF/HFmrEF.**
(DOCX)

## Acknowledgments

The authors are grateful to all the physicians and nurses taking care of our patients with heart failure, all the laboratory assistants analysing the blood samples, and all the surgeons performing the heart transplants and LVAD implantations.

## Author Contributions

**Conceptualization:** Petr Lokaj, Jindrich Spinar, Lenka Spinarova, Filip Malek, Jan Krejci, Petr Ostadal, Tomas Ondrus, Jiri Parenica.

**Data curation:** Klara Benesova, Jiri Jarkovsky.

**Formal analysis:** Klara Benesova, Jiri Jarkovsky.

**Funding acquisition:** Jiri Parenica.

**Investigation:** Petr Lokaj, Jindrich Spinar, Lenka Spinarova, Filip Malek, Ondrej Ludka, Jan Krejci, Petr Ostadal, Dagmar Vondrakova, Karel Labr, Monika Spinarova, Monika Pavkova Goldbergova, Marie Miklikova, Katerina Helanova, Ilona Parenicova, Vladimir Jakubo, Roman Miklik, Tomas Ondrus, Jiri Parenica.

**Methodology:** Petr Lokaj, Jindrich Spinar, Lenka Spinarova, Filip Malek, Jan Krejci, Petr Ostadal, Tomas Ondrus, Jiri Parenica.

**Project administration:** Petr Lokaj, Tomas Ondrus, Jiri Parenica.

**Resources:** Petr Lokaj, Ondrej Ludka, Dagmar Vondrakova, Karel Labr, Monika Spinarova, Monika Pavkova Goldbergova, Marie Miklikova, Katerina Helanova, Ilona Parenicova, Vladimir Jakubo, Tomas Ondrus, Jiri Parenica.

**Software:** Klara Benesova, Jiri Jarkovsky.

**Supervision:** Petr Lokaj, Tomas Ondrus, Jiri Parenica.

**Validation:** Petr Lokaj, Klara Benesova, Jiri Jarkovsky, Tomas Ondrus, Jiri Parenica.

**Visualization:** Petr Lokaj, Tomas Ondrus, Jiri Parenica.

**Writing – original draft:** Petr Lokaj, Tomas Ondrus, Jiri Parenica.

**Writing – review & editing:** Petr Lokaj, Roman Miklik, Tomas Ondrus, Jiri Parenica.

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
