## [Decision Letter · Decision Letter 0]

25 Feb 2021

PONE-D-21-01502

Prognostic value of high-sensitivity cardiac troponin I in heart failure patients with mid-range and reduced ejection fraction

PLOS ONE

Dear Dr. Ondrus,

Thank you for submitting your manuscript to PLOS ONE. After careful consideration, we feel that it has merit but does not fully meet PLOS ONE’s publication criteria as it currently stands. Therefore, we invite you to submit a revised version of the manuscript that addresses the points raised during the review process.

Two Reviewers have raised some concerns regarding Your manuscript that need to be addressed.

We look forward to receiving your revised manuscript.

Kind regards,

Giuseppe Rengo, M.D., Ph.D.

Academic Editor

PLOS ONE

Additional Editor Comments (if provided):

Your manuscript can not be accepted in the present form. The Reviewers raised some concerns that need to be addressed.

Journal Requirements:

2) We note that the grant information you provided in the ‘Funding Information’ and ‘Financial Disclosure’ sections do not match.

3) Please include captions for all your Supporting Information files at the end of your manuscript, and update any in-text citations to match accordingly. Please see our Supporting Information guidelines for more information: http://journals.plos.org/plosone/s/supporting-information.

Reviewers' comments:

Reviewer's Responses to Questions

**Comments to the Author**

1. Is the manuscript technically sound, and do the data support the conclusions?

Reviewer #1: Yes

Reviewer #2: Yes

2. Has the statistical analysis been performed appropriately and rigorously? 

Reviewer #1: Yes

Reviewer #2: Yes

3. Have the authors made all data underlying the findings in their manuscript fully available?

Reviewer #1: Yes

Reviewer #2: Yes

4. Is the manuscript presented in an intelligible fashion and written in standard English?

Reviewer #1: Yes

Reviewer #2: Yes

5. Review Comments to the Author

Reviewer #1: In this study, the authors examined the association of hs-cTnI and prognosis in 520 patients hospitalized with stable chronic heart failure. The main findings were that hs-cTnI correlated with age, LVEF, and renal function, and they correlate with the NT-proBNP levels in study patients. The value of hs-cTnI also identify the patients that are at a higher risk of a combined endpoint. Those data are felt to be interesting. However, this paper might be improved if the authors could reconsider the following points.

Major Comments

#1 The authors should show in the definition of chronic heart failure.

Because there is no mention of who made the diagnosis or the echocardiography results made the diagnosis. Were these patient's enrolled the A&E department, cardiology clinic, medical team etc? What is the accuracy of diagnosis? This is the most important potential bias in the study.

#2 The authors how to think about the timing of the measurement of hs-cTnI for their conclusion. Anytime is OK?

#3 The authors should show the figure of ROC curves, because they determine the cut off level of hs-cTnI from the ROC curve. The readers probably confirm the difference of cut off level of hs-cTnI affect the prognosis. The cutoff level of hs-cTnI >17ng is not a strong predictor because the median hs-cTnI level is 17 ng in the group of "Without endpoint"

Other comments

#4 The reviewer cannot understand why it is divided the patients into HFrEF and HFmrEF in this study. Is the prognosis different between HFrEF and HFmrEF? Has the prognostic factor or troponin I cutoff changed?

Reviewer #2: Petr et al. performed a very interesting study on the prognostic value of high-sensitivity troponin I in patients with HfmrEF and HfmrEF.

The authors found s-cTnI level to be an independent marker for increased risk of an adverse prognosis.

Although limited by the exclusion of HfrEF population, the paper is well-written.

I have just several further comments to improve its quality:

1. The end point was reached only in 14% of patients (73 out of 520 patients). Perhaps it would have been compelling to have a longer follow-up and to consider hospitalizations as secondary end-point too.

2. The authors generated a ROC curve of the troponin value (17 ng / L), whose patients above are at high risk. However, the population is not homogeneous (80% men), too many in NYHA 2 class (about 60%). My concerns is that this representative sample does not reflect the actual real-world HF patient population

3. It might be intriguing to separate and compare patients with HfmrEF and HfrEF in two cohorts to show the difference between the two.

6. PLOS authors have the option to publish the peer review history of their article (what does this mean?). If published, this will include your full peer review and any attached files.

Reviewer #1: No

Reviewer #2: No

---

## [Author Response · Author response to Decision Letter 0]

3 Apr 2021

Please, see attached document with responds

---

## [Decision Letter · Decision Letter 1]

30 Apr 2021

PONE-D-21-01502R1

Prognostic value of high-sensitivity cardiac troponin I in heart failure patients with mid-range and reduced ejection fraction

PLOS ONE

Dear Dr. Ondrus,

Thank you for submitting your manuscript to PLOS ONE. After careful consideration, we feel that it has merit but does not fully meet PLOS ONE’s publication criteria as it currently stands. Therefore, we invite you to submit a revised version of the manuscript that addresses the points raised during the review process.

The manuscript has been significantly improved. As requested by the Reviewer no. 2, please, provide data on the hospitalization rate.

We look forward to receiving your revised manuscript.

Kind regards,

Giuseppe Rengo, M.D., Ph.D.

Academic Editor

PLOS ONE

Journal Requirements:

Additional Editor Comments (if provided):

Reviewer #2 requests to add hospitalization as secondary endpoint in order to improve the overall impact of the study.

Reviewers' comments:

Reviewer's Responses to Questions

**Comments to the Author**

1. If the authors have adequately addressed your comments raised in a previous round of review and you feel that this manuscript is now acceptable for publication, you may indicate that here to bypass the “Comments to the Author” section, enter your conflict of interest statement in the “Confidential to Editor” section, and submit your "Accept" recommendation.

Reviewer #1: All comments have been addressed

Reviewer #2: All comments have been addressed

2. Is the manuscript technically sound, and do the data support the conclusions?

Reviewer #1: Yes

Reviewer #2: Yes

3. Has the statistical analysis been performed appropriately and rigorously? 

Reviewer #1: Yes

Reviewer #2: Yes

4. Have the authors made all data underlying the findings in their manuscript fully available?

Reviewer #1: Yes

Reviewer #2: (No Response)

5. Is the manuscript presented in an intelligible fashion and written in standard English?

Reviewer #1: Yes

Reviewer #2: Yes

6. Review Comments to the Author

Reviewer #1: The Authors have done well to address the question of the reviewer's comments. Overall this is a good topic and is novel.

Reviewer #2: The paper has improved and it's almost suitable for publication.

However, I still believe that adding hospitalization as secondary end-point would be an interesting data for readers.

7. PLOS authors have the option to publish the peer review history of their article (what does this mean?). If published, this will include your full peer review and any attached files.

Reviewer #1: No

Reviewer #2: No

---

## [Author Response · Author response to Decision Letter 1]

13 Jun 2021

Please, see attached file Respons to reviewers

---

## [Decision Letter · Decision Letter 2]

14 Jul 2021

Prognostic value of high-sensitivity cardiac troponin I in heart failure patients with mid-range and reduced ejection fraction

PONE-D-21-01502R2

Dear Dr. Ondrus,

We’re pleased to inform you that your manuscript has been judged scientifically suitable for publication and will be formally accepted for publication once it meets all outstanding technical requirements.

Kind regards,

Giuseppe Rengo, M.D., Ph.D.

Academic Editor

PLOS ONE

Additional Editor Comments (optional):

Reviewers' comments:

Reviewer's Responses to Questions

**Comments to the Author**

1. If the authors have adequately addressed your comments raised in a previous round of review and you feel that this manuscript is now acceptable for publication, you may indicate that here to bypass the “Comments to the Author” section, enter your conflict of interest statement in the “Confidential to Editor” section, and submit your "Accept" recommendation.

Reviewer #2: All comments have been addressed

2. Is the manuscript technically sound, and do the data support the conclusions?

Reviewer #2: Yes

3. Has the statistical analysis been performed appropriately and rigorously? 

Reviewer #2: Yes

4. Have the authors made all data underlying the findings in their manuscript fully available?

Reviewer #2: Yes

5. Is the manuscript presented in an intelligible fashion and written in standard English?

Reviewer #2: Yes

6. Review Comments to the Author

Reviewer #2: Thank you to the authors for addressing my request of adding hospitalization data. The paper is now suitable for publication. Well done!

7. PLOS authors have the option to publish the peer review history of their article (what does this mean?). If published, this will include your full peer review and any attached files.

Reviewer #2: **Yes: **Alberto M. Marra

---

## [Editor Report · Acceptance letter]

22 Jul 2021

PONE-D-21-01502R2 

Prognostic value of high-sensitivity cardiac troponin I in heart failure patients with mid-range and reduced ejection fraction 

Dear Dr. Ondrus:

I'm pleased to inform you that your manuscript has been deemed suitable for publication in PLOS ONE. Congratulations! Your manuscript is now with our production department. 

Kind regards, 

on behalf of

Dr. Giuseppe Rengo 

Academic Editor

PLOS ONE